# Mitochondrial Transfer into Human Oocytes Improved Embryo Quality and Clinical Outcomes in Recurrent Pregnancy Failure Cases

**DOI:** 10.3390/ijms24032738

**Published:** 2023-02-01

**Authors:** Yoshiharu Morimoto, Udayanga Sanath Kankanam Gamage, Takayuki Yamochi, Noriatsu Saeki, Naoharu Morimoto, Masaya Yamanaka, Akiko Koike, Yuki Miyamoto, Kumiko Tanaka, Aisaku Fukuda, Shu Hashimoto, Ryuzo Yanagimachi

**Affiliations:** 1Department of Obstetrics and Gynecology, HORAC Grand Front Osaka Clinic, Osaka 530-0011, Japan; 2Department of Research, HORAC Grand Front Osaka Clinic, Osaka 530-0011, Japan; 3Reproductive Science Institute, Graduate School of Medicine, Osaka Metropolitan University, Osaka 545-8585, Japan; 4Department of Obstetrics and Gynecology, Nippon Life Hospital, Osaka 550-0006, Japan; 5Department of Obstetrics and Gynecology, IVF Namba Clinic, Osaka 550-0015, Japan; 6Department of Research, IVF Namba Clinic, Osaka 550-0015, Japan; 7Department of Reproductive Technology, HORAC Grand Front Osaka Clinic, Osaka 530-0011, Japan; 8Department of Integrated Medicine, HORAC Grand Front Osaka Clinic, Osaka 530-0011, Japan; 9Department of Obstetrics and Gynecology, IVF Osaka Clinic, Osaka 577-0012, Japan; 10Department of Anatomy, Biochemistry and Physiology, John A. Burns School of Medicine, University of Hawaii, Honolulu, HI 96822, USA

**Keywords:** mitochondrial transfer, embryo quality, clinical outcome, cognitive development, mtDNA

## Abstract

One of the most critical issues to be solved in reproductive medicine is the treatment of patients with multiple failures of assisted reproductive treatment caused by low-quality embryos. This study investigated whether mitochondrial transfer to human oocytes improves embryo quality and provides subsequent acceptable clinical results and normality to children born due to the use of this technology. We transferred autologous mitochondria extracted from oogonia stem cells to mature oocytes with sperm at the time of intracytoplasmic sperm injection in 52 patients with recurrent failures (average 5.3 times). We assessed embryo quality using the following three methods: good-quality embryo rates, transferable embryo rates, and a novel embryo-scoring system (embryo quality score; EQS) in 33 patients who meet the preset inclusion criteria for analysis. We also evaluated the clinical outcomes of the in vitro fertilization and development of children born using this technology and compared the mtDNA sequences of the children and their mothers. The good-quality embryo rates, transferable embryo rates, and EQS significantly increased after mitochondrial transfer and resulted in 13 babies born in normal conditions. The mtDNA sequences were almost identical to the respective maternal sequences at the 83 major sites examined. Mitochondrial transfer into human oocytes is an effective clinical option to enhance embryo quality in recurrent in vitro fertilization-failure cases.

## 1. Introduction

It is likely that there are multiple causes of repeated failures in assisted reproduction [1]. Patients bear a heavy burden and waste time through recurrent failures caused by deteriorated oocyte quality (maturation and fertilization failure), embryo quality (cleavage, blastulation, and hatching failure), and implantation failure. In particular, the decline in oocyte and embryo quality is an unsolvable issue [2,3]. Clinicians who treat patients with poor oocyte quality make various attempts to rescue them. They may change the ovarian stimulation protocol in an assisted reproductive technology program, or they instruct their patients to change their lifestyles such as sleep or meal patterns. The physicians may recommend increasing physical exercise. However, these attempts do not work well in most cases, or they take a long time to be effective.

In terms of cell biology, the low-quality potential of oocytes is attributable to the mutation of nuclear DNA, defects in chromatid separation, chromosome decondensation, and spindle detachment [4]. Nevertheless, the genetic, metabolic, and molecular dysfunction in ooplasm where mitochondria play a main role cannot be disregarded.

Recently developed cell engineering technologies could be one of the solutions for this issue. Among them, mitochondria technology could be a breakthrough in surmounting the problem.

Mitochondria are powerhouses that provide energy, and the dysfunction of the organelle may disrupt ooplasmic maturation and oogenesis. The inability of mitochondria to provide sufficient energy to oocytes has long been recognized as a significant cause of low quality in oocytes and embryos [5].

The exclusive maternal transmission of mitochondria DNA (mtDNA) tends to induce incurable physical and genetic disorders, known as mitochondrial disease. In research on these diseases, several therapeutic approaches have been described, including pronuclear transfer [6], spindle transfer [7], and polar body transfer [8]. These technologies in the field of cell engineering could be applied to patients with poor-quality oocytes. However, they cannot be used in clinical practice because of low success rates and ethical issues.

In reproductive medicine, improving oocyte viability using mitochondrial transfer (MT) has been attempted in cattle and other mammals [9]. In the past, ooplasmic transfer was attempted to improve oocyte viability; however, the procedure was suspended because of the high risk of heteroplasmy [10]. Additionally, the idea of transferring mitochondria from healthy fully grown oocytes [5], or other somatic cells, such as granulosa cells, was proposed and attempted [11,12,13]. However, these methods were not approved as conventional clinical technology for ethical reasons and low efficacy.

Putative oogonial stem cells (p-OSCs) in the adult ovary are excellent sources of mitochondria for supplementing oocytes because of the lower mtDNA mutations and deletions in these cells compared with most other cells [14,15,16]. Moreover, using of mitochondria from autologous germline stem cells is an option to avoid heteroplasmy. There have been reports of MT using mitochondria from oogonia stem cells [15,17,18,19]; however, to our knowledge, they did not refer to the status of children born with this technology.

Here, we transferred autologous mitochondria extracted from oogonia stem cells into mature oocytes of the patients with recurrent failures of assisted reproductive treatment and investigated whether mitochondrial transfer improves the quality of embryos in the patients. Subsequently, we indicate the clinical outcomes and the prognostic follow-up data of babies born after this procedure, which validated the improvement in oocyte quality by the technology.

## 2. Results

### 2.1. Cell Preparation: p-OSCs Were Successfully Isolated from Ovarian Cortical Tissues

To isolate functional mitochondria from patients undergoing MT, we first collected and characterized p-OSCs obtained from each patient’s ovarian cortical tissue. Assuming that all harvested tissue specimens were cuboid, the mean volume of the three harvested ovarian pieces of cortical tissue from each patient was 245.6 ± 91.4 mm^3^. The total number of p-OSCs collected from each patient (*n* = 52) ranged from 11,327 to 664,000 (mean: 172,778 ± 115,782). The average density of the p-OSCs in the ovarian cortical tissue was 708.1 ± 426.4/mm^3^. The p-OSCs were mainly round or oval (Figure 1a) and showed cell-membrane and cytoplasmic expression of DDX4 on fluorescence micrographs (Figure 1b–d). Each p-OSC had a large nucleus surrounded by a small amount of cytoplasm (Figure 1e, N). Oval (most; Figure 1f, empty arrowheads) and elongated (rare; Figure 1f, filled arrowhead) mitochondria with cristae were scattered in the cytoplasm (Figure 1e–g). Thus, we successfully isolated and characterized the p-OSCs from the ovarian cortical tissue of the patients undergoing MT.

### 2.2. Clinical Outcomes

A total of 52 women aged 27–49 years underwent MT (Table 1). We simultaneously injected spermatozoa and mitochondria into each oocyte (702 mature oocytes from 52 patients). The fertilization rate was 61.5%, and the clinical pregnancy rate per transfer was 23.8%. We had 11 live births (3 sets of twins and 8 singletons), 1 other intrauterine fetal death (1 fetus of a twin pregnancy), and 4 miscarriages. The average implantation rate was 18.6%, and the average live birth rate was 17.5%.

The clinical outcomes of the two groups, Pre-MT (previous cycles before MT) and MT (the cycle of MT), were compared using data on the embryos from previous treatment cycles from 33 patients (Table 2). The patients’ demographic data and clinical results are summarized in Table 2. There were no differences in oxidative stress (d-ROMs) or biological antioxidant potential (BAP). The average transferable embryo rates per zygote significantly increased from 33.2 ± 23.2 to 62.4 ± 22.3, and the average good-quality embryo rate per zygote significantly increased from 6.9 ± 9.3 to 23.7 ± 24.4 after MT. The average EQS of the transferable embryos on Day 3 significantly increased from 1.1 ± 1.0 to 1.9 ± 0.9. The clinical pregnancy and live birth rates per embryo transfer improved from 0% to 27.4% and from 0% to 21.5% after MT, respectively.

### 2.3. Analysis of mtDNA Sequences of the Patients (Mothers) and Their Children Born after MT

For the validation of the efficacy of the MT, we tested whether the autologous mitochondrial transfer bore any genetic risk in mtDNA by assessing the mtDNA sequence-concordance rate between four mothers and their children (five children, including one set of twins) (Table 3). We compared the nucleotide sequences of 83 major sites in the mtDNA extracted from the blood samples between the mothers and their children. The mtDNA sequences were 100% identical for four children and 99% identical in one child when compared to the respective maternal sequences at the 83 major sites examined.

### 2.4. Follow-Up of Children Born after MT

The most effective method to prove the efficacy of MT is the confirmation of a child’s normality. None of the 13 children born after MT had recognizable anomalies, such as external malformations. When the six children reached 18 months old or older, their motor and cognitive development were investigated. Their average body weights and heights were 2892.9 ± 164.7 g and 49.1 ± 1.2 cm, respectively, at birth and 10,257.1 ± 755.2 g and 80.8 ± 2.4 cm, respectively, at 18 months of age. They were within the normal ranges. Pediatric examinations of appearance, respiration, skin color, muscle strength, and neural reflexes revealed no abnormalities in any of the six children’s physical development.

The scores of the six children in the Kinder Infant Development Scale (KIDS), which was used to evaluate cognitive development, were 95.2 ± 11.4 for the physical motor subscale, 114.7 ± 11.8 for the manipulation subscale, 114.6 ± 20.9 for the receptive language subscale, 98.9 ± 16.9 for the expressive language subscale, 99.4 ± 13.0 for the language concept subscale, 102.4 ± 13.3 for the social relationships with children subscale, 124.7 ± 27.4 for the social relationships with adults subscale, 116.2 ± 17.1 for the discipline subscale, 111.2 ± 6.3 for the feeding subscale, and 109.6 ± 10.6 overall (Figure 2). As a result, all six children had normal motor and cognitive development.

## 3. Discussion

Oxidative stress is known to relate to mitochondrial function in oocytes. Mitochondria produce reactive oxygen species as electrons leak from the electron-transport chain, which deteriorates mtDNA and their metabolic competence. We performed d-ROMs tests to evaluate the overall oxidative status of the patients. In this study, the patients who underwent MT did not have high systemic oxidative stress levels. Serum antioxidant capacity and concentrations of oxidative stress markers did not show oxidative stress in oocytes. Therefore, we conclude from our result that the dysfunction of oocyte mitochondria cannot be inferred from those systemic values. Additionally, the oxidative stress levels in the serum might differ from those in the ovaries because of the differences in their vascularity.

There is no fixed evidence that mitochondrial function recovers through the reduction in oxidative stress. Our previous study [20] showed that there was no elevation of reactive oxygen species after mitochondria transfer.

In the clinical setting, infertility professionals meet patients with repeated IVF failure, where the poor quality of oocytes is responsible. Poor-quality oocytes do not mature, fertilize, cleavage, or blastulate. Nucleic and cytoplasmic factors are responsible for this issue, and mitochondria play a key role in the latter. A shortage of provided ATP generated by mitochondria causes detrimental conditions in oogenesis and embryogenesis.

It is not easy to assess embryo quality. In clinical settings, the evaluated embryos need to be vital and to maintain their homeostasis. Several noninvasive assessments have been conducted. Traditional methods for cleaved embryos [21] and blastocysts [22] have been widely applied for their simple usability. Alpha scientists in reproductive medicine and the special interest group in European Society of the Human Reproduction and Embryology proposed the Istanbul consensus [23]. This scoring system is designed to assess oocytes, gametes, cleaved embryos, morula, and blastocysts separately by considering phenomena in embryology, such as the time from insemination and multinucleation. This method is sophisticated; however, it does not include the element of pregnancy, which is the essential outcome for embryos. Therefore, we developed a new embryo quality system (EQS) [24]. This scoring system includes pregnancy rates and enables us to score the cleaved embryo and blastocysts on the same platform. The clearest method to assess embryo quality is to evaluate good-quality embryo rates and transferable embryo rates. In this study, the quality was improved in all three embryo assessment methods.

Embryos should be essentially assessed not only in terms of their morphology but also from multiple standpoints, including metabolic and genetic status; however, this is not possible in noninvasive assessments. Therefore, we indicated the clinical outcomes, children’s development, and incidence of heteroplasmy as validations of embryo homeostasis after MT. The fact that the clinical outcomes were improved and the children born with this technology showed normal development indirectly indicates that the MT improved the oocyte quality.

As further confirmation of the improvement in oocytes, we conducted a genetic analysis. We analyzed the mtDNA sequences of babies born after MT and their mothers to determine whether there were any heteroplasmy-related genetic problems, which would have been revealed in the mitochondrial sequencing results. It should be noted that mutations and deletions are liable to occur in mtDNA because of the absence of histones, which protect other genetic material. The mitochondrial genome is hypermutable compared with nuclear DNA, and in heteroplasmy, mutated mtDNA can coexist as a subset of total cellular mtDNA [25]. A common deletion, ΔmtDNA^4977^, has been reported in 33% of oocytes and 8% of embryos in humans [26]. There is concern that mutations or deletions may manifest mitochondria-induced phenotypes in children; however, the mutant mtDNA must typically exceed 70% of a heteroplasmy threshold level to result in functional defects [27].

The mechanism of mtDNA deletion has yet to be elucidated. The factors involved could include replication errors or breaks in double-stranded DNA [28]. Mitochondria have a built-in control system, called the bottleneck phenomenon and mitophagy [29,30]. Through these mechanisms, most malfunctioning mitochondria are eliminated, and the remaining normal mitochondria proliferate [30]. Phenotypic manifestations of genetic defects only occur when a threshold level of mutation is exceeded, termed “the mitochondrial threshold effect” [31]. Therefore, even if the pathogenic mutation of mtDNA increases during mitochondrial transfer, mitochondrial disorders are unlikely to be expressed phenotypically. Although the mtDNA bottleneck function can significantly prevent the transmission of detrimental mtDNA mutations to the next generation, its efficacy still needs to be confirmed. The mtDNA data obtained in this study confirmed that the tertiary mitochondrial source used during the MT process, namely, autologous mitochondria, did not transmit any significant mtDNA-inheritance-related defects to the recipients and that the maternal mtDNA was transmitted dominantly to the children. The gene concordance rate between the mother and her child(ren) in the mitochondrial transfer cases was higher than that in the reported cases in the non-transfer group [32]. Our observations indicate that no significant genetic abnormalities related to the transplantation of autologous mitochondria into oocytes have arisen up to this point.

Recently, intracellular MT has been studied in the fields of the central nervous [33], respiratory [34]**,** and cardiovascular [35] systems. In these broad fields, the transferred mitochondria were shown to play a role in tissue repair. Referring to the knowledge of intracellularly transferred mitochondria, oocytes could be reconstructed with their original qualities by mitochondria supplementation.

As options for enhancing embryo quality, various MTs have been applied in different manners. Ooplasmic transfer from young donors was an easy way to achieve this purpose and successfully yielded over 30 babies [10]. However, this method had issues that required discussion. Other functional organelles might be involved in the content of ooplasm, and the method created concern about heteroplasmy and the destruction of donor oocytes. An early stage trial of MT was reported in 1998 [36]. Using the compartmentalization of mouse and human oocyte mitochondria in the ooplasm, the study demonstrated an apparent increase in ATP production in the oocytes and the persistence of activity in the transferred mitochondria. The mitochondria sourced from granulosa cells were transferred to bovine [12] and human oocytes [13], and both studies indicated the effect on MT. However, this method might raise concerns because the mitochondria in the granulosa cells are different in their morphologies.

Some studies reported the use of mitochondria from stem cells in animals. The mitochondria in mature oocytes are oval-shaped, and well-developed cristae rarely appear. The characteristics of mitochondria from stem cells, such as embryonic stem cells and induced pluripotent stem (iPS) cells, resemble those of oocytes [37,38]. A study of the transfer of iPS-derived mitochondria indicated that their morphological characteristics and membrane potential are similar to those of oocytes and rescued the impaired developmental potential of embryos in aging female mice [39]. Umbilical cord mesenchymal stem cells have also been adopted [40]. Therefore, using stem cells as a source for mitochondria harvest is promising for the clinical application of MT.

We used mitochondria harvested from oogonia stem cells. Several studies have confirmed the presence of stem cells (i.e., p-OSCs) in the ovarian cortex [41,42,43,44,45,46] that can be isolated by taking advantage of the cell-membranous protein expression, DDX4 [41,47,48]. However, there are contradictory opinions on the existence of p-OSCs, insisting that DDX4+ cells are somatic cells [49,50]. The mitochondria of young and healthy somatic cells are known to have fewer mutations and mtDNA deletions [51,52] than those of older cells. Therefore, the stem cells in the ovarian cortex make it possible to extract potentially healthy tissue-specific autologous mitochondria that are free of cumulative damage to their genomes and do not result in heteroplasmy or aneuploidy [14,15,16,41]. On the other hand, there is a study that presents results that are opposite to ours [18]. Considering that the blastulation rates of objective patients in the study were much higher than ours, there is a difference in patient backgrounds between the two studies.

One of the critical factors relating to low oocyte quality is aging. The recent increase in advanced-aged patients had led to an increase in low-quality oocytes and embryos, an issue to be discussed in clinical practice. The mtDNA mutation and deletion relate to the diminished oocyte quality [36]. Mitochondria quality and quantity decline upon aging. Age-related defect of mitochondria biogenesis induces abnormalities in mitochondrial respiratory chain proteins and uncoupling of the respiratory chains, causing aneuploidy in the nucleus and follicle development arrest [53]. Aging is not the main subject of this research; however, MT may rejuvenate oocytes and solve the issue.

Adipose-tissue-derived stem cells are promising candidates for use as autologous mitochondria sources for MT. Successful MT in mice has been reported [54]. The study noting such results showed that the mitochondria in adipose-tissue-derived stem cells expressed normal morphology and increased oocyte quality in aged mice. The morphologies of mitochondria in adipose stem cells are similar to those of mature oocytes [20]. In addition, MT using these cells increased ATP levels in murine oocytes and improved blastulation rates while reactive oxygen species levels remained unchanged [20]. The clinical issue with MT using the mitochondria in oogonia stem cells is the high cost due to the need for laparoscopy. In contrast, harvesting adipose tissue is much more accessible and affordable for patients.

## 4. Materials and Methods

### 4.1. Study Design and Patient Selection

Fifty-two patients (27–49 years old) underwent MT at the HORAC Grand Front Osaka Clinic, Osaka, Japan, between February 2016 and February 2019 after providing written informed consent. This study was registered on 1 April 2016, at the University Hospital Medical Information Network Center Clinical Trials Registry (No. UMIN000021). Infertility was diagnosed following the American Society for Reproductive Medicine and Japan Society of Obstetrics and Gynecology guidelines. The patients included in this comparative analysis had previously experienced multiple (mean 5.13) failures of in vitro fertilization (IVF) or intracytoplasmic sperm injection (ICSI) because the quality of embryos was poorly defined as approximately 80% of embryos with arrested development at the 4- or 8-cell stage or because their embryos were highly fragmented (60% of entire blastomeres). The exclusion criteria for this study were the presence of uterine structural anomalies, polycystic ovaries, or premature ovarian failure, as these factors may lead to female infertility, which requires clinical approaches other than enhancing oocyte quality.

Of the 52 patients who underwent MT, 33 conformed to the preset inclusion criteria to enable the evaluation, had data on the embryos from their previous treatment cycles that were precisely logged, and were under 42 years old. The demographic characteristics of age, body mass index, and anti-Müllerian hormone were compared between the two groups, comprising patients before and after the MT. The embryo quality was compared before and after the MT as follows: (1) Good-quality embryo rates were calculated by conventional morphological methods. Grade 1 and 2 embryos on Day 2 and Day 3 by Veeck’s criteria and blastocysts better than 4BB by Gardner’s criteria were defined as good-quality embryos. (2) Transferable embryo rates were calculated in each period. The transferable embryos were defined as follows: Cleavage-stage embryos better than grade 3 according to Veeck’s criteria [21] and blastocyst-stage embryos better than grade 3 without a grade C inner cell mass according to Gardner’s criteria [22] were transferable. (3) Comparison used with a new embryo scoring system (EQS) [24]. This scoring system was newly developed based on clinical pregnancy rates for each embryo; it offers an advantage in scoring the viability of developing fetuses compared with the scoring system that has been conventionally used. Furthermore, it has a feature that helps to evaluate embryos at any stage on the same platform.

### 4.2. Evaluation of the Oxidative Stress Status of Patients

As reactive oxygen species affect mitochondrial function [55,56], we investigated the oxidative stress status of the patients included in this study. To evaluate oxidative stress and antioxidative potential of patient blood plasma, we used Diacron reactive oxygen metabolites (d-ROMs) and biological antioxidant potential (BAP) tests with a Free Radical Elective Evaluator (Diacron International SRL, Grosseto, Italy). Blood samples were collected from the patients during their first visit to the HORAC Grand Front Osaka Clinic. The d-ROM and BAP tests were conducted according to the manufacturer’s instructions.

### 4.3. Isolation of p-OSCs, Extraction of Mitochondria from p-OSCs, and Injection of Mitochondria into Oocytes (Figure 3)

The p-OSCs were isolated according to a previously described protocol [57]. In brief, three pieces of tissue (approximately 6 × 6 × 1 mm) were collected by laparoscopy from the ovarian cortex of each patient. These tissue samples were cryopreserved using a programmed cryopreservation method at −1 °C/min from 18 °C to −80 °C using a Crysalys Embryo Freezer (PTC-9500, Biogenics Inc., Napa, CA, USA) and stored in liquid nitrogen until needed. Frozen cells were thawed in a water bath at 37 °C, and a cell suspension was prepared by mechanical and enzymatic dissociation of the tissue using gentleMACS™ Octo Dissociator (Miltenyi Biotech, Bergisch Gladbach, Germany). The cells were stained with fluorescein-labeled monoclonal or polyclonal anti DEAD Box Helicase 4 (DDX4), VASA antibodies (OvaScience, Inc., Waltham, MA, USA), or Bioss Antibodies, (catalog no. bs-3597R-A488, Beijing, China), and p-OSCs were isolated using fluorescence-activated cell sorting on a Sony SH800 cell sorter (Sony Biotechnology Inc., San Jose, CA, USA). The cells were stored in liquid nitrogen after freezing in a Crysalys Embryo Freezer using the aforementioned program. For immunofluorescence analysis, the sorted cells were counterstained with 4′,6-diamidino-2-phenylindole and dihydrochloride (DAPI, Dojindo Chemical Research Institute, Kumamoto, Japan) and analyzed using Keyence fluorescence microscopy (BZ-X800, Keyence Corp., Osaka, Japan). The quantity and density of p-OSCs (numbers per mm^3^) were recorded.

Mitochondria were extracted from p-OSCs on the day of fertilization via ICSI according to the patented protocol [15]. The isolation and quantification of mitochondria from p-OSC were performed following a previously described protocol, with minor modifications [10,15,58], to attain an established minimum threshold number of mitochondria (500 mtDNA copies) to improve ICSI results [59]. In brief, p-OSCs were thawed and washed in Hanks’ balanced salt solution (Gibco-14175-095; Thermo Fisher Scientific, Waltham, MA, USA) and then resuspended in a standard mitochondria isolation buffer (420 mM D mannitol, 140 mM sucrose, 10 mM HEPES, 2 mM KCl, and 2 mM egtazic acid at pH 7.2) with 0.5% human serum albumin (catalog No. 9988, Irvine Scientific, Santa Ana, CA, USA). Next, the cells were homogenized for 10 min at 5 °C (Sorvall Legend ×1, Thermo Fisher Scientific) by repeated infusion and withdrawal using a syringe pump connected to a 25 G cannula (Harvard Apparatus, PHD Ultra 703007, BioSurplus, Inc., San Diego, CA, USA). The homogenates were centrifuged at 800× *g* for 10 min before the supernatant was carefully collected. Next, the supernatant was centrifuged at 7000× *g* for 30 min at 5 °C (Sorvall Legend ×1, Thermo Fisher Scientific) to separate mitochondria from smaller cell debris. After the supernatant was discarded, mitochondrial pellets were kept in respiration buffer (225 mM D mannitol, 200 mM sucrose, 10 mM KCl, 10 mM Tris-HCl, and 5 mM KH2PO4) at pH 7.2 with 0.5% human serum albumin until use. Immediately before ICSI, 2 μL of mitochondrial suspension was loaded into an 80-micrometer Flexipet pipette (Cook Medical LLC, Bloomington, IN, USA) and centrifuged at 10,000× *g* at 4 °C for 20 min (Sorvall Legend Micro 21R, Thermo Fisher Scientific). After centrifugation, the concentrated mitochondria and supernatant were placed separately in the ICSI dish (OOSAFE^®^ ICSI/IMSI dish, SparMED ApS, Farum, Denmark). The protocol for loading mitochondria and spermatozoa into an injection pipette is shown in Figure 3. Approximately 2 pL of mitochondrial suspension and a spermatozoon were injected together into an oocyte. The injection was performed within 2 h after mitochondrial isolation. Because mitochondria and spermatozoa were adherent, this procedure was difficult even for well-trained embryologists.

**Figure 3 ijms-24-02738-f003:**
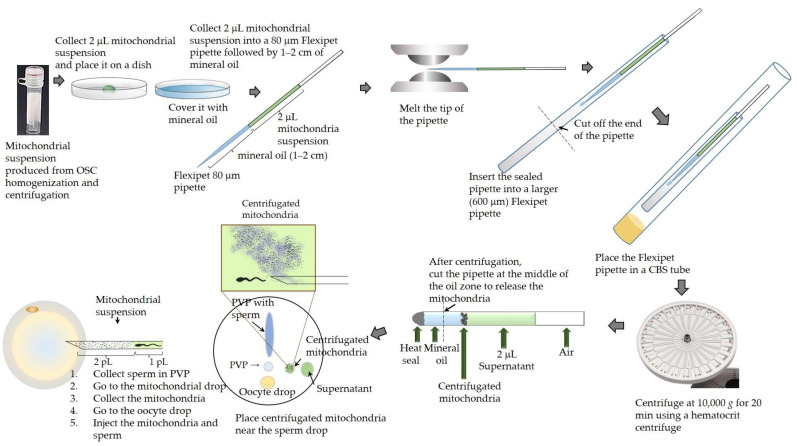
The method for isolation of p-OSCs, extraction of mitochondria, and injection of mitochondria into oocytes.

### 4.4. Ovarian Stimulation and Oocyte Retrieval

The ovarian stimulation protocol was selected based on anti-Müllerian hormone levels, the antral follicle counts, and the number of oocytes retrieved in previous cycles. A long gonadotropin-releasing hormone antagonist protocol [60] was used for patients with a standard or high ovarian reserve. No or mild stimulation with clomiphene citrate/letrozole and/or follicle-stimulating hormone /human chorionic gonadotrophin (hCG) [61] was used for patients with a low ovarian reserve. Oocytes were retrieved 36 h after hCG injection (5000 IU) when follicles reached more than 18 mm in diameter.

### 4.5. Embryo Transfer and Diagnosis of Pregnancy

Embryos were cultured and transferred to the patients on days 2, 3, and 5 after the day of ICSI. The fresh or cryopreserved and thawed embryos were transferred onto the endometrium prepared by routine estrogen and progesterone substitution therapy. Serum β-hCG levels were determined 12 days after embryo transfer to test for pregnancy. Clinical pregnancy was confirmed by the presence of a gestational sac detected using ultrasonography at 6 weeks of gestation.

### 4.6. Transmission Electron Microscopic Examination of p-OSCs

The p-OSCs were fixed in cold 3% glutaraldehyde in cacodylate buffer for 1 h followed by postfixation in 1% aqueous osmium tetroxide for 1 h. They were rapidly dehydrated in graded ethanol solutions with a final rinse in acetone and embedded in Epon. Semithin sections were stained with toluidine blue. Ultrathin sections were stained with uranyl acetate and lead citrate before examination under a transmission electron microscope (H-7600, Hitachi Ltd., Tokyo, Japan).

### 4.7. Comparison of mtDNA Sequences of Children and Their Mothers

#### 4.7.1. DNA Extraction and Polymerase Chain Reaction (PCR) Amplification

Peripheral blood samples were collected from four women who agreed to provide blood samples for mtDNA analysis and from their five children born after MT (one woman had twins). DNA was extracted using PureLink™ (Thermo Fisher Scientific) following the manufacturer’s protocol. We performed two long-range PCRs with two overlapping primer sets (Fragment A: mt.10796–3370, Fragment B: mt.2817–11590) to generate two large DNA products (9143 bp and 8773 bp) suitable for combined and uniform fragmentation.

The mtDNA was amplified using the oligonucleotide primer sets 5′-CCACTGACATGACTTTCCAA-3′ (A_F10796) and 5′-AGAATTTTTCGTTCGGTAAG-3′ (A_R3370) for Fragment A, and 5′-GCGACCTCGGAGCAGAAC-3′ (B_F2817) and 5′-GTAGGCAGATGGAGCTTGTTAT-3′ (B_R11590) for Fragment B, as previously reported, using a Veriti™ thermal cycler (Thermo Fisher Scientific).

The thermocycling conditions were as follows: denaturation at 98 °C for 30 s, followed by 25 cycles at 98 °C for 10 s, 63 °C for 10 s (Fragment A) or 57 °C for 10 s (Fragment B), and 72 °C for 4 min; a final extension at 72 °C for 5 min. PCR products were purified using AMPure XP beads (Beckman Coulter Inc., La Brea, CA, USA). PCR purification was performed according to the manufacturer’s instructions.

#### 4.7.2. Library Construction

Library construction involved enzymatic shearing, adapter ligation, and size selection, all according to the manufacturer’s instructions. Both fragments were normalized to a quantity of 100 ng and then pooled. Amplicons were enzymatically sheared into suitably sized fragments using an Ion Xpress™ Plus Fragment Library Kit (Thermo Fisher Scientific). For barcoded libraries, an Ion Xpress™ P1 Adapter and Ion Xpress™ Barcode Adapter (Thermo Fisher Scientific) were used to sequence multiple samples simultaneously. DNA fragments of approximately 480 bp from the fragmented and adapter-ligated libraries were selected using E-Gel™ SizeSelect™ II Agarose Gels, 2% (Thermo Fisher Scientific). The quantity of the size-selected library was determined through a real-time PCR approach using an Ion Library TaqMan Quantitation Kit (Thermo Fisher Scientific) and the template dilution factor was calculated to obtain a final concentration of 100 pM per target.

#### 4.7.3. Template Preparation and Sequencing

For template preparation, targets were subjected to emulsion PCR and clonal amplification using Ion 510™, Ion 520™, and Ion 530™ Kit-Chef (Thermo Fisher Scientific) for 400 base libraries. The prepared templates were subsequently loaded onto Ion 520™ chips (Thermo Fisher Scientific) using an Ion Chef™. Single-read sequencing was performed on a high-throughput semiconductor sequencing platform on the Ion GeneStudio™ S5 Prime System (Thermo Fisher Scientific). All protocols followed the manufacturer’s instructions.

#### 4.7.4. Data Analysis

Following the successful sequencing of mitochondrial libraries, read trimming, base calling, and mapping to the reference mitochondrial genome, the Revised Cambridge Reference Sequence was completed using Ion Torrent Suite Software version 5.12.1; Thermo Fisher Scientific). The alignment was done using the Torrent Mapping Alignment Program versions 5.12-27 and 5.12-28 downloaded from https://github.com/iontorrent/TS/tree/master/Analysis/TMAP (the date of analysis: 25 March 2020). Variant calling was performed using the Torrent Variant Caller plugin (version 5.12-28) with default settings.

### 4.8. Follow-Up of Children Born after MT

The children were examined once at 18, 21, 23, or 24 months after MT by a clinical psychologist and a pediatrician. Six children born after MT were subjected to physical examination (i.e., appearance, respiration, skin color, muscle strength, and neural reflexes) by a pediatrician. Furthermore, their motor and cognitive development was assessed by a clinical psychologist using the Type B test (age for 12–35 months) of the Kinder Infant Development Scale (KIDS) (Center of Developmental Education and Research, Tokyo, Japan). At the time of this psychological test, the six children were aged 21 months (female), 23 months (female), 24 months (male), 24 months (female), 18 months (male), and 24 months (male). The KIDS comprises a list of behaviors in the following nine subscales: physical motor (25 behaviors), manipulation (25 behaviors), receptive language (13 behaviors), expressive language (13 behaviors), language concepts (13 behaviors), social relationships with children (13 behaviors), social relationships with adults (13 behaviors), discipline (14 behaviors), and feeding (13 behaviors).

The developmental quotient (DQ) is a number representing the developmental status of a child and is determined by dividing the KIDS by the child’s chronological age and by multiplying this number by 100; thus, a DQ of 100 indicates the appropriate development for the child’s chronological age. Significant developmental delay is defined as a DQ of 70 or less [62,63]. The children were assessed for DQ by their mothers and a psychologist who examined their behaviors in each subscale [64]. Additionally, the total DQ of each child was assessed.

### 4.9. Statistical Analysis

For intergroup comparisons, the Kruskal–Wallis one-way analysis of variance or chi-square test was used. Variables are presented as arithmetic means ± standard deviations in each group. Additionally, for the fertilization rate and embryo development rate analysis between the pre-MT and MT, we used the repeated measures analysis of covariance using Statistical Package for the Social Sciences (IBM Corp, Armonk, NY, USA). All other statistical analyses were performed using Prism 5 (GraphPad Software Inc., San Diego, CA, USA). Any *p*-values of less than 0.05 were used to indicate statistical significance.

## 5. Conclusions

Improving embryo quality is a critical issue for patients who experience repeated IVF failures. We validated embryo quality using three methods and found that the embryo quality was improved after MT. Subsequently, we established that this technology aided in the birth of 13 healthy babies whose physical and cognitive development was normal. Furthermore, no evidence of mtDNA heteroplasmy was found in these children. Therefore, we concluded that MT technology is a beneficial clinical option to improve oocyte quality and the subsequent clinical results for patients with recurrent failures. The limitation of this study is its retrospective nature. To evaluate the efficacy of new technology, a prospective randomized study is ideal. However, this was not possible because the object population for this investigation was patients who were in a desperate situation and did not accept being enrolled in an experimental study. The future scope of our project is to provide technically easier and inexpensive procedures that can be used broadly. MT sourced from adipose stem cells meets this requirement.

## Figures and Tables

**Figure 1 ijms-24-02738-f001:**
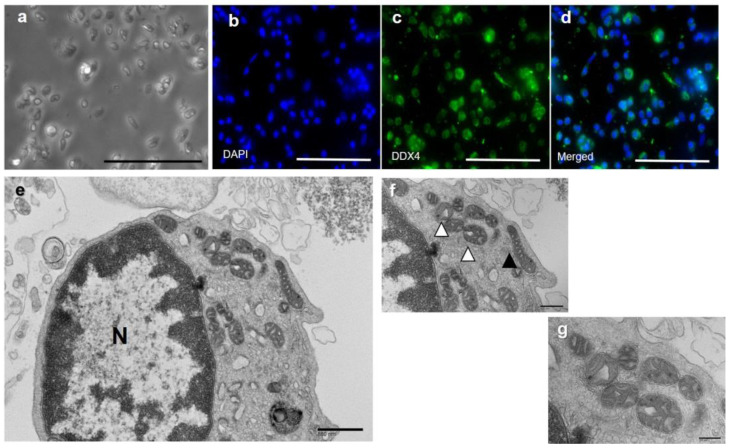
Morphology of OSCs. (**a**) Phase-contrast micrograph. (**b**–**d**) Immunofluorescence staining of DDX4 in sorted oogonial stem cells (OSCs; different cells from (**a**). Ovarian cortex cells were stained with a DDX4 antibody, sorted, and then fixed for imaging. (**c**) The fluorescence microscopy images show the cell membranous and cytoplasmic expression of DDX4. DAPI was used as the counterstain (**b**: DAPI), and no nuclear expression of DDX4 was observed (**d**: merged) (**a**–**d** scale bar: 100 μm). (**e**–**g**) Transmission electron micrograph of a human oogonial stem cell. (**e**) Electron micrograph of an OSC. The cell contains a large nucleus (N, ×25,000, scale bar: 800 nm). (**f**) Mitochondria at high magnification. Oval (empty arrowheads) and elongated (filled arrowheads) mitochondria are shown (×50,000, scale bar: 400 nm). (**g**) Cristae are clearly identified (×100,000, scale bar: 200 nm).

**Figure 2 ijms-24-02738-f002:**
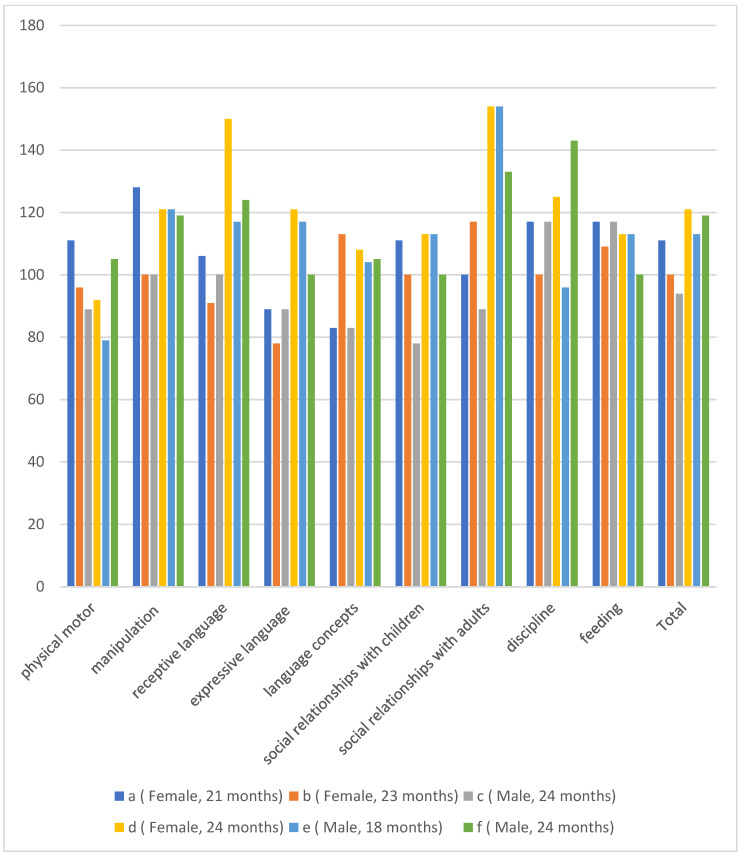
Results of KIDS test. The motor and cognitive development scores of the children born via MT were assessed with the Type B KIDS (Kinder Infant Developmental Scale) test. KIDS comprises a list of behaviors in the nine subscales. The average scores of each subscale ranged from 95.2 to 124.7, and the scores were not under cut-off range of 70. Therefore, all children developed normally.

**Table 1 ijms-24-02738-t001:** Summary of ICSI for all 52 patients who received mitochondrial transfer.

No. of Retrieved Oocytes	884	No. of Miscarriages	4
No. of oocyte retrieval procedures	105	% Miscarriage	26.7 (4/15)
No. of mature oocytes	702	No. of ectopic pregnancies	1
Average no. of mature oocytes per oocyte retrieval cycle	8.4 (884/105)	No. of twin pregnancies	3
No. of fertilized oocytes	432	No. of implantations	18
% Fertilization	61.5 (432/702)	% Implantation	18.6 (18/97)
No. of embryo transfers	63	No. of Live births	11
No. of transferred embryos	97	% Live birth	17.5 (11/63)
No. of clinical pregnancies	15	No. of babies born	13
% Clinical pregnancies per embryo transfer	23.8 (15/63)		

The data from all 52 patients who underwent MT. We had 11 live births (3 sets of twins and 8 singletons). One twin baby died in the uterus. Therefore, 13 babies were born after the treatment.

**Table 2 ijms-24-02738-t002:** Comparison of demographic data, oxidative stress, and embryo quality between Pre-MT and MT groups, and clinical outcomes.

	Pre-MT Group	MT Group	*p* Value
Number of patients	33	33	
Age	34.6 ± 3.7	36.4 ± 3.9	n.s.
Body mass index (BMI)	21.1 ± 3.2	21.2 ± 3.3	n.s.
Anti-Müllerian hormone (AMH) (ng/mL)	4.2 ± 4.8	1.8 ± 1.5	n.s.
Diacron-Reactive Oxygen Metabolites tests (d-ROM) (CARR U)	354.9 ± 66.1	321.1 ± 81.5	n.s.
Biological Antioxidant Potential (BAP) (μmol/L)	2222.5 ± 54.9	2270.3 ± 207.6	n.s.
Total oocyte No.	718	387	
Average oocyte No. retrieved per patient	8.4 ± 4.4	11.4 ± 6.9	*p* < 0.05
Total fertilized oocytes No.	459	208	
Average No. of fertilized oocytes per patient	5.2 ± 3.1	6.2 ± 3.5	*p* < 0.05
Average fertilization rate (%)	61.8 ± 19.9	59.3 ± 22.8	n.s.
Total No. of transferable embryos	163	119	
Average transferable embryo rate per zygote (%)	33.2 ± 23.2	62.4 ± 22.3	*p* < 0.05
Average good-quality embryo rate per zygote (%)	6.9 ± 9.3	23.7 ± 24.4	*p* < 0.05
Average EQS of the transferable embryo on Day 3	1.1± 1.0	1.9 ± 0.9	*p* < 0.05
No. of embryos transferred	98	78	
No. of embryo transfers	91	51	
Clinical pregnancy rate per embryo transfer (%)	0	27.4 (14/51)	*p* < 0.05
Live birth rate per embryo transfer (%)	0	21.5 (11/51)	*p* < 0.05
No. of babies born	0	13	
Miscarriage rate (%)	ND	21.4 (3 /14)	

There were no significant differences in demographic data and oxidative status between the Pre-MT and MT groups. The d-Rom value can be broken down as follows: {Pre-MT < 340 (CARR U) (57.1%), 341–400 (19%), 401–500 (19%), 500 (4.7%)}, {MT < 340 (CARR U) (69.5%), 341–400 (17.4%), 401–500 (8%), 500 (4.3%)}. The good-quality embryo rates, transferable embryo rates per zygote, and EQS significantly increased. The clinical pregnancy rates and live birth rates were improved after MT. EQS: Embryo Quality Scoring. *p* < 0.05 statistically significant.

**Table 3 ijms-24-02738-t003:** Comparison of mtDNA in mothers and their children born after MT.

	Comparison	Concordance Rate
Family A	Mother vis-à-vis child	100%
Family B	Mother vis-à-vis child	100%
Family C	Mother vis-à-vis child	100%
Family D *	Mother vis-à-vis child 1	99%
Family D *	Mother vis-à-vis child 2	100%

Peripheral blood samples were collected from four female patients and their five children born after MT. Eighty-three major sites in the mtDNA reference sequences were compared. There was a difference between the mother and her one child in Family D at only one site. * The mother in Family D had non-identical twins.

## Data Availability

Not applicable.

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
