# Peer review of "Mitochondrial Transfer into Human Oocytes Improved Embryo Quality and Clinical Outcomes in Recurrent Pregnancy Failure Cases"

_ijms, 2023, doi:10.3390/ijms24032738_

Round 1

Reviewer 1 Report

The following should be noted and corrected accordingly:

1. How practicable is your proposed model in real-time?

2. Is it cost-efficient?

3. Some diagrams and terms are not properly explained

4. Grammar requires minor re-editing

5. Are the numbers and formulas here generic or generated by you?

6. What are the limitations of the study?

7. The Introduction is too brief. It should contain the main objectives and organization of the paper

8. There is a notable lack of a Literature Survey/Related Works section

9. What is the future scope of the study?

10. Study and consider the following related papers to embellish your paper:

• https://doi.org/10.3390/diagnostics12112643

• https://doi.org/10.3390/s22031076

• https://doi.org/10.3390/s22155574

Reviewer 2 Report

I recommend that they discuss their method in the introduction. Why was mitochondrial transfer chosen? Was a reduced number of mitochondria or impairment of mitochondrial function observed in the pre-MT group?

pp. 4.

 There were no differences in oxidative stress (d-ROMs) or antioxidant potential (BAP).

corrected: . There were no differences in oxidative stress (d-ROMs) or biological antioxidant potential (BAP).

Table 2.

Biological Antioxidant Potential (BAP) (mmol/L)

corrected: Biological Antioxidant Potential (BAP) (micromol/L)

How did the authors calculate the Average fertilization rate?

It would be more informative to present ranges with the number of cases than the average value of d-ROMs. eg: <340 (CARR U) (n or %), 341-400 (n, %), 401-500 (n, %), 500 (n, %)

2.3.

What sequences were tested when examining the identity of maternal and child mtDNA sequences?

The missing reference must be replaced.

4.3.

In brief, three pieces of tissue (approximately 6 × 6 × 1 mm3 ) were collected by laparoscopy from the ovarian cortex of each patient.  Did you mean mm?

The first paragraph of the discussion (Oxidative stress is known to... ... differ from those in the ovaries because of the differences in their vascularity) is not relevant, I recommend its omission. Based on the BAP and ROM averages, I do not think that the women included in the study have significant oxidative stress (see my suggestion for Table 2.).  The oocyte is a tiny part of the human mass. Serum antioxidant capacity and concentrations of oxidative stress markers do not show oxidative stress in oocytes. Dysfunction of oocyte mitochondria cannot be inferred from these values either. The authors also describe this in the last sentence.

Reviewer 3 Report

Regarding the manuscript entitled “Mitochondrial Transfer into Human Oocytes Improved Embryo Quality and Clinical Outcomes in Recurrent Pregnancy Failure Cases”, the authors aimed to investigate whether mitochondrial supplementation improves human embryo quality and provides subsequent acceptable clinical results and normality to children born due to the technology.

The author contributed well and the quality of the manuscript is up to the mark. The abstract is comprehensively written. The introduction started from very basic and continued till elaborating and explaining the problem well. The material and methods section was written in a way that the research conducted could easily be replicated and provide guidance to the future researcher. The research conducted and the write-up of the article are outstanding. the results and discussion are well elaborated. I studied it in detail and realized its importance, and I recommend it for publication after English editing.

Please avoid using “we”

Author Response

Thank you very much for your kind words. Your comments encourage us to go forward to save infertility patients who are in big trouble. We have put our revised manuscript to English editing again.

Reviewer 4 Report

The manuscript “Mitochondrial transfer into human oocytes improved embryo quality and clinical outcomes in recurrent pregnancy failure cases” suggests a potential predictor for a patient with female infertility. Interestingly, the authors conducted new approaches to mitochondria transfer and suggest a new possibility for infertility treatment. Thus, the authors need to describe the possibility of this diagnostic method in practice.

Some points have to be corrected.

1. It is necessary to describe the improvement rate of oocyte quality by mitochondrial transfer more concretely. Because, overall, the difference in standard error is large. Especially the fertility rate and good quality of embryos.

2. Adipocyte mitochondria benefits have been found. Is there a high possibility of applying this method to infertility treatment? How many clinics actually apply to mitochondrial transfer? Is there any problem with safety?

3. Is there any change in the energy of the oocyte before and after mitochondrial transfer?

Also, does inducing a decrease in mitochondrial activity in egg cause aging?

4. Will mitochondria recover when reducing oxidative stress? When mitochondria produce ATP, a large amount of active oxygen is also produced. Please describe the balance of energy and oxidative stress.

Round 2

Reviewer 1 Report

No further recommendations 

Reviewer 4 Report

 I think that the revised manuscript has been fundamentally improved and that it includes the contents requested by the referees and editorial team.